# Know Pain, No Pain? Preliminary Testing and Application of a New Tool to Assess Biopsychosocial Pain Concepts in Children

**DOI:** 10.3390/children10050814

**Published:** 2023-04-29

**Authors:** Linda Wickering, Catherina Lautwein, Hanna Nitsche, Michael Schneider, Tanja Hechler

**Affiliations:** 1Department of Clinical Psychology and Psychotherapy for Children and Adolescents, University of Trier, 54296 Trier, Germany; s1liwick@uni-trier.de (L.W.);; 2Department of Educational Psychology, University of Trier, 54296 Trier, Germany; 3Department of Clinical Psychology for Children and Adolescents, University of Münster, 48149 Münster, Germany

**Keywords:** chronic pain, pediatric pain, biopsychosocial, pain concept, pain science education, questionnaire, pain treatment, cognitive interview

## Abstract

To deliver tailored pain science education, assessing children’s biopsychosocial pain concepts is necessary. As validated tools are lacking, a new tool is presented, the biopsychosocial pain concept matrix (BiPS matrix), which assesses children’s biological, psychological, and social pain concepts in five domains according to the Common-Sense Model of Self-Regulation (Hagger and Orbell, 2003): (1) illness identity, (2) causes, (3) consequences, (4) duration, and (5) treatment. The present preliminary study aims to (1) assess the items’ readability and understandability in cognitive interviews with *N* = 9 healthy children (9 to 19 years, *M* = 13.78 years, *SD* = 3.05; 44% female) and (2) pre-test the BiPS matrix within an online survey of *N* = 27 healthy children (9 to 19 years, *M* = 13.76 years, *SD* = 3.03; 56% female). Results revealed difficulties in understanding some items. Children’s understanding increased with age. Age, chronic pain status, and pain in the social environment were positively associated with the BiPS total score, whereas the latter explained the most variance in pain concepts of children. Patient-focused methods such as cognitive interviews proved essential in testing the readability and understanding of items in children. Future studies are warranted to further validate the BiPS matrix.

## 1. Introduction

Chronic pain is a relevant health problem in children, with 5.1% of children between the ages of 8–16 years suffering from moderate to severe chronic pain [1]. Chronic pain can severely impact different aspects of life: children suffering from chronic pain indicate a worse quality of life and, in particular, those with high levels of pain-related disability report greater school absences than children without chronic pain problems.

The biopsychosocial model provides an explanatory approach to explain the development and maintenance of chronic pain. It incorporates biological, psychological, and social aspects of pain, emphasizes their importance [2,3,4], and is the basis for diagnostic and therapeutic decisions [2,3].

Pain science education is a key intervention in pain treatment in both children and adults [4], but its mechanisms and effectiveness, particularly for youth, are hardly understood [5]. In pain science education, patients learn about the underlying biopsychosocial mechanisms of pain [6]. It is an approach aiming to not only challenge preexisting beliefs, such as those focusing on a biomedical explanation for pain, but also to introduce pain as a biopsychosocial phenomenon. Some of these knowledge acquisition processes can be described as enrichment, where new knowledge is added to a learner’s knowledge base. However, sometimes the knowledge to be learned might be incompatible with preexisting knowledge structures in a learner’s long-term memory. In this case, learning requires conceptual change, that is, a restructuring of the knowledge in long-term memory [7]. Conceptual knowledge is notoriously difficult to measure, partly because conceptual knowledge is multidimensional and heterogeneous, sometimes even fragmented, in its nature. Therefore, high internal consistencies and high Cronbach’s alpha values of conceptual knowledge scales cannot be expected [8,9]. This highlights the importance of a multidimensional approach for operationalizing pain concepts.

Pain concepts play an important role in pain science education [10,11,12]. A patient’s pain concept refers to “how they understand what pain actually is, what function pain serves, and what biological processes are thought to underpin it” [11]. Misconceptions such as “If you have a slipped disc, you must have surgery", on the other hand, differ from currently accepted concepts. They frequently interfere with learning and have explanatory power but compete effectively with correct scientific concepts [13]. Depending on their pain concept, patients forecast the course of their disease and adjust their management of pain accordingly [14,15], e.g., avoiding movement and activity or seeking solely medical care based on a biological pain concept [16,17]. Different studies could show that adult patients with chronic pain often have a biological concept of their pain, e.g., conceptualizing tissue damage as the underlying single biological cause (e.g., [7,18,19]). Having a biological pain concept can then lead to increased resting to not take a physical toll on hurting body parts, consequently causing even more pain in the future.

Misconceptions regarding pain can lead to a vicious cycle of maintaining chronic pain due to inadequate coping strategies, resulting in patients’ prolonged suffering. Therefore, it is inevitable to uncover and eventually change misconceptions in patients through pain science education to treat them successfully [10,19].

Pain concepts are not only based on biopsychosocial pain knowledge but also on people’s illness perception during their disease. People generate mental representations of their illnesses by interpreting available information. According to the *Common-Sense Model of Self-Regulation* [20], these mental representations can be subdivided into five domains: (1) illness identity, (2) cause, (3) consequences, (4) duration/timeline beliefs, and (5) possible treatments and action plans. In their meta-analysis comprising *N* = 45 empirical studies mostly focusing on adults, these domains could be shown in adults suffering from >20 different illnesses, e.g., acute pain, multiple sclerosis, and osteoarthritis [14], demonstrating the stability of the domains.

Assessing concepts of pain is important for the conceptualization of pain science education [21]. Pain concepts of adults can be assessed by using specific tools such as the Neurophysiology of Pain Questionnaire (NPQ) [22,23], which mainly covers knowledge about biological processes. Other relevant questionnaires for adults are the Survey of Pain Attitudes (SOPA) [24], the Pain Beliefs Questionnaire (PBQ) [25], and the Pain Belief Screening Instrument (PBSI) [26]. Significantly fewer tools exist for children and adolescents. The Pain Perception Questionnaire for Young People (PPQ-YP) is designed for 10- to 16-year-old children but was tested exclusively in adolescents with secondary chronic pain due to juvenile arthritis [15]. The Conceptualization of Pain Questionnaire (COPAQ) [27] includes 15 items for 8- to 17-year-old children to assess their concepts of pain. The Concept of Pain Inventory (COPI) by Pate et al. [28] includes 14 items for an age range between 8 and 12 years and covers biological and psychological aspects of chronic pain. However, neither the COPAQ nor the COPI covers social aspects on a separate subscale. The biopsychosocial perspective is therefore neglected. 

To our knowledge, no multidimensional tool assesses all three aspects of pain, i.e., biological, psychological, and social, on separate subscales while covering all five domains postulated by the Common-Sense Model of Self-Regulation [29].

This preliminary study presents the first data on children’s biopsychosocial pain concepts using a new tool, the Biopsychosocial Pain Concept Matrix (BiPS matrix) [30]. The BiPS matrix assesses children’s pain concepts in five domains, with biological, psychological, and social items for each domain according to the five domains of illness representations [14]: (1) cause, (2) consequences, (3) type of pain, (4) timeline, and (5) treatment. It therefore has an underlying multidimensional matrix structure with several items in each cell (Appendix A).

The questionnaire was initially developed in 2019 and tested in a sample of *N* = 47 healthy adults (*M* = 33.55 years, *SD* = 14.84) [31]. Afterwards, the tool was presented to interdisciplinary experts who rated items on relevance and usefulness [30]. The experts attested to the high relevance and usefulness of the BiPS matrix, and experts’ comments were used to further reduce and adapt items for use in children and adolescents. The present version consists of 40 items.

The purposes of the present project were (1) to assess children’s reading problems, their difficulties in explaining meaning, and unknown words by use of cognitive interviews; and (2) to pre-test the BiPS matrix within an online sample to gain an initial and preliminary insight into modulating factors of a child’s concept of pain, such as age, chronic pain status, prior pain knowledge, and pain in the social environment.

## 2. Materials and Methods

The BiPS matrix is a multidimensional tool assessing pain concepts [5]. Two versions are available (for children and adults) in German. The children’s version was derived from a larger pool of 72 items which was generated and further developed with insight from pain experts in previous research [30]. The goal is to assess a child’s concept of pain with a multidimensional tool covering biological, psychological, and social aspects of pain while also including the five domains of illness representations [14]. It therefore has an underlying multidimensional matrix structure with several items in each cell (Appendix A). An example of an item assigned to psychological aspects of pain and causes of pain is “*Thoughts can influence pain intensity*”. The version used in this study contains 40 items. Each item is rated on a 5-point Likert scale with responses ranging from “*totally disagree*”, “*agree*”, “*I’m not sure*”, “*disagree*” to “*totally agree*”. A total score is calculated with higher scores representing pain concepts more closely intertwined with contemporary pain science. The same version of the questionnaire was used in part 1 (cognitive interviews) and part 2 (online study). The children’s parents provided informed consent for participation in the study.


**Part 1: Cognitive Interviews**


Participants were informed about the study in person. Cognitive interviews were conducted in December 2020 and January 2021 with a convenience sample of children in the circle of acquaintances of the researchers. Children were asked if they wanted to participate in the study. Parents provided consent. Two researchers [LW, HN] conducted the interviews: 5 interviews were conducted online via Zoom (due to the COVID-19 pandemic) and 4 were conducted in person. A total of 9 children aged 9–19 years (*M* = 13.78, *SD* = 3.05; 55.5% male) completed the study. Interviews lasted between 30 and 55 min (*M* = 44.44; *SD* = 3.59). The interview guideline is included in Appendix B.

We included healthy children aged 9–19 years with sufficient knowledge of the German language and sufficient reading abilities, and we excluded children younger than 9 years of age or older than 19 years with insufficient knowledge of the German language and insufficient reading abilities, similar to the procedures in other studies using cognitive interviews [28]. Interviews were conducted according to interview guidelines [32]. At the beginning of the cognitive interview, the interviewers clarified that the interview was not about achieving a specific grade and that there were no right or wrong answers. The researchers then explained the aim of the study, i.e., to get the participants’ opinion on items to improve the BiPS matrix for use in this age group. Interview questions were aligned with a guideline used in former studies testing a similar instrument, the COPI [28]. Children were handed a list of all items and then asked to (1) read each item out loud, (2) explain the meaning of each item in their own words, (3) tell if the sentence made sense to them, (4) name unknown words, and, finally, (5) make suggestions for rewording. Participants were encouraged to add further comments and suggestions at the end of the interview. The interview guideline is provided in Appendix B.

**Rating:** Interviewers wrote down and rated all answers in a protocol. Interviewers rated “*yes*” if no reading problems occurred (fluent reading, no pauses between words, words are pronounced correctly). Interviewers also rated “*yes*” if the explained meaning of items was in line with the scientific meaning of items. Whether the item made sense was answered by the participants and then written down in the protocol by the interviewers. Unknown words were also written down, as were any suggestions for rewording. The duration of the interviews varied between 30 and 55 min. 

**Quantitative data analysis:** Children’s reading problems and correct explanations of items were summarized by count for each item, and unknown words were written down and then counted. Children’s rating of “*items making sense*” was also counted. 

**Qualitative data analysis:** Suggestions for rewording were written down and categorized for qualitative analysis and discussion in the research group, and, ultimately, for future adaptation of the instrument.


**Part 2: Online Study**


Participants were informed about the study via email or in person at a local, cooperating support center for children of parents with cancer (*Papillon Trier*), enabling us to test this tool in a sample of children, who had some degree of experience with chronic pain (through their parents). Data was collected using online assessment via the online software *unipark* (www.unipark.de). The same inclusion and exclusion criteria as in the cognitive interviews in part 1 of the study were used. Eligible children and their parents were informed about the research project and invited to participate in the study via email. A link was sent, granting immediate access to the questionnaire; a QR code was also provided. Participation in the study was voluntary; no monetary incentives were given. All information on the study objectives, data use, and storage were given orally as well as written in *unipark*. All parents gave consent via *unipark*. The online study was conducted between December 2020 and January 2021.

**Measurement:** The online questionnaire included (1) sociodemographic items such as gender, age, and type of school; (2) *chronic pain status* (“*Do you suffer from chronic pain?*”) If children indicated that they suffered from chronic pain, detailed questions regarding their pain characteristics were asked [33]; (3) items to asses *prior pain knowledge* (self-evaluation of pain knowledge on a 5-point Likert scale); and (4) *pain in the social environment* (“*Do you know anyone suffering from chronic pain?*” on a 5-point Likert scale ranging from “*no*” to “*yes*”, *and “I engage in this topic*”).

**BiPS matrix:** The version of the BiPS matrix used in this study contains 40 items that children answered on a 5-point Likert scale ranging from “*totally disagree*” to “*totally agree*”, with the middle category being “*I’m not sure*” (similar to the procedure in part 1). After each item, participants were asked to indicate whether they understood the item (tick “*yes*” or “*no*”). Participants were encouraged to add comments after each item, which was voluntary.

**Data pre-processing.** The data of the online survey was downloaded from *unipark* into the Statistical Package for the Social Studies (SPSS), version 24 (IBM, Armonk, NY, USA) for further analyses. New variables were computed to calculate means for the biological, psychological, and social subscale and the five domains, as well a total score of the BiPS matrix. After each item, participants were asked to indicate “*yes*” or “*no*”, depending on whether they understood the item. From this, a comprehension score was computed (percentage of items understood). Comments were collected and analyzed qualitatively. 

**Data analysis:** All data were screened for normality, appropriate ranges, and outliers before further analyses. Descriptive statistics such as means, standard deviations, and minimum and maximum values were computed. Conceptual pain knowledge was assessed via the BiPS total score, and response patterns were investigated by tallying answers (“*agree*”, “*disagree*”, “*unsure*”) and proportion calculations. Item-total correlations, internal consistency, and item difficulty were computed for preliminary testing of psychometric characteristics. Before the regression analyses, Pearson Product–Moment correlations between age and the BiPS total score were computed. Due to non-parametric variables, Spearman’s rank correlations were computed to test the influence of *chronic pain status*, *prior pain knowledge*, and *pain in the social environment* on the BiPS total score as well as on all separate dimensions. Those variables displaying significant relationships were then included in the regression analyses.

We performed a series of stepwise multiple regression analyses with age (in years), *chronic pain status* (0 = no pain, 1 = pain), *prior pain knowledge* (1 = very little, 2 = little, 3 = a bit, 4 = a lot, 5 = plenty) and *pain in the social environment* (1 = I don’t know anyone suffering from chronic pain, 2 = not sure, 3 = yes, but I don’t witness it, 4 = yes and I witness it, 5 = yes and I engage in this topic) on (1) the BiPS total score, (2) the biological subscale, (3) the psychological subscale, and (4) the social subscale. All assumptions were met: The data was checked for linearity between predictors and outcome variables; no outliers were found. Autocorrelation was checked via the Durbin–Watson statistic; no multicollinearity was detected, and the residuals showed homogeneity of variance. A significance level of *p* < 0.05 was applied, and the effect size *R*^2^ is reported.

Statistical analyses were computed using the Statistical Package for Social Sciences (SPSS, version 25).

## 3. Results

### 3.1. Part 1: Cognitive Interviews

**Reading problems:** Children could read out loud 21 out of the 40 items (52.5%) without any reading problems. Difficulties arose with items including unknown words such as “chronic” (items 1, 22, 23, 24, 25, 30, 33, 34, 38, 39), “warning message” (item 3), “pain intensity” (items 6, 7, 8, 10, 40) or “danger detector” (item 1). Minor reading problems occurred, such as small breaks, leaving out a word, or floundering. The questionnaire is currently only available in German but can be provided upon request.**Explain meaning:** Children (at least *N =* 1 out of a total *N* = 9) struggled with explaining the meaning of the content of 26 of the 40 items (65%) (items 1–8, 10, 1–14, 20, 22–25, 27–29, 30, 33, 34, 37, 38, 40). Items were rated problematic as soon as at least one child explained the meaning of an item incorrectly. A total of 38 out of 40 items (95%) were explained correctly by more than 50% of children. No problems arose in 14 of the 40 items (35%) (items 9, 11, 15–19, 21, 26, 31, 32, 35, 36, 39). At least three out of nine children struggled with the following items: 1, 2, 4, 7, 8, 13, 20, 22, and 28. A salient finding was that some children explained items according to real-life experiences which did not match the intended meaning (Table 1). Problems explaining the meaning co-occurred with unknown words such as “chronic”. Children explained some items using emotions or by describing content as “injuries to the soul”.**Do items make sense?** All children rated 17 of the 40 items (42.5%) as making sense to them. Again, in this calculation, items were rated as “*not making sense*” as soon as at least one participant rated an item as not making sense. Ten items were rated as not making sense by one child. Eight items were rated as not making sense by two children and five items were rated as not making sense by three children. That means that no more than three children per item rated that specific item as not making sense. In total, no items were rated as making sense by less than 66% of the children. Often, children rated items as not useful without giving a specific reason.**Unknown words:** Unknown words were “chronic” (unknown in 5/9), “acute” (4/9), “warning function” (3/9), “therapist” (3/9), and “protective function” (2/9). One child also did not know the following words: “spinal cord”, “warning message”, “danger message”, “details”, “pain intensity”, “culture”, and “existing”.**Suggestions for rewording:** Most children (*N =* 7) suggested explaining difficult words at the beginning of the questionnaire or paraphrasing them with easier known words. Sometimes concrete suggestions for rephrasing were made. Suggestions included facilitating sentence structure, subdividing really long items into two items, as well as reducing the item number (in particular, deleting items with similar meanings).

### 3.2. Part 2: Online Study

**Characteristics of the online sample.** A total of *N* = 27 children aged 9–19 years (*M* = 13.76, *SD* = 3.03; 44.4% male) completed the study. The average duration for the online questionnaire lasted between 7 and 36 min (*M* = 13 min). Children attended grades 4 to 13. Eight participants (29,6%) suffered from chronic pain (headache, *N =* 4; stomachache, *N =* 2; back pain, *N =* 1; pain in extremities, *N =* 1). Children rated their mean *pain intensity* with *M* = 4 (*SD* = 2.45) on the NRS 0–10. For the following analyses, *N* = 26 participants were considered, and one participant was excluded as an outlier in the BiPS total score.

**Comprehension.** The percentage of items understood was good in this sample, with a comprehension score of *M* = 94.17% (*SD* = 12.69). The comprehension score was not normally distributed (Shapiro–Wilk test, *p* < 0.001). The high score could be explained by a ceiling effect (16 subjects had no comprehension problems, and 6 other subjects indicated only one problem). In total, 19 comments were made. Comments concerned semantic comprehension and content of items such as “*I don’t understand how friends and pain should be associated*”. Comments also revealed unknown words such as “chronic”, “acute”, “warning function”, “protection”.

**Descriptive statistics of the BiPS matrix.** The total score in the BiPS matrix was *M* = 2.70 (*SD* = 0.29, *range* = 1.75–3.13). The BiPS total score was normally distributed. 

**Item and scale analysis of the BiPS matrix.** To test items and psychometric characteristics of the questionnaire, such as the item-total correlation of all dimensions and difficulty index, psychometric properties of the vertical dimensions (biological, psychological, and social aspects of chronic pain) and horizontal dimensions (the five domains of illness representations) of the BiPS matrix were calculated (Appendix C).

After the stepwise elimination of nine items (1, 4, 10, 13, 18, 25, 27, 30, 34) due to negative item-total correlation, the total BiPS score for the remaining 31 items was *M* = 2.80 (*SD* = 0.39, range 0–4), with a minimum of 1.90 (range 1–4) and a maximum of 3.58 (range 1–4). The total score of the BiPS matrix showed high internal consistency (*α* = 0.86) and moderate difficulty (*P_i_* = 67.4). Psychometric properties of the vertical and horizontal dimensions of the adjusted BiPS matrix are presented in Table 2. All scales were normally distributed (*p* > 0.05); only the horizontal dimension *timeline* was not. As expected, (see introduction), some Cronbach’s alpha values were smaller than 0.7. This is a necessary consequence of the heterogeneous nature of conceptual knowledge and cannot be interpreted as indicating low reliability [8,9]. Therefore, we deem all alpha values acceptable in our case. 


**Modulating factors on the BiPS matrix: correlational analyses**


Appendix D depicts the bivariate relationships between children’s pain characteristics (age, chronic pain status, prior pain knowledge, and pain in the social environment) and the children’s concept of pain (BiPS total score, biological, psychological, and social subscales).

**Age:** There was a significant positive relationship between age and the BiPS total score (*r* = 0.400, *p* < 0.05) as well as between age and the psychological (*r* = 0.475, *p* < 0.05) and social subscales (*r* = 0.442, *p* < 0.05).**Chronic Pain Status of Children:** Chronic pain status of children was assessed through self-evaluation (“*yes*” or “*no*”). There was a significant positive relationship between the chronic pain status of children and the BiPS total score (*r* = 0.464, *p* < 0.05) as well as between chronic pain status and the psychological (*r* = 0.491, *p* < 0.01) and social subscales (*r* = 0.461, *p* < 0.05).**Prior pain knowledge:** Mean *prior pain knowledge* of children (self-evaluation of pain knowledge on a 5-point Likert scale) was *M* = 3.35 (*SD* = 0.94, *range* = 1.75–3.13). Prior pain knowledge was unrelated to the BiPS total score and all subscales.**Pain in the social environment:** Pain in the children’s social environment was assessed through self-evaluation on a 5-point Likert scale (“*Do you know people suffering from chronic pain?*”) and showed significant positive correlations with the BiPS total score (*r* = 0.62; *p* < 0.01) as well as with the psychological (*r* = 0.683; *p* < 0.01) and social subscales (*r* = 0.437; *p* < 0.05).


**Modulating factors on the BiPS matrix: Results of the multiple regression analyses to predict pain concepts in children**


**BiPS total score:** Based on significant correlations, age, chronic pain status, and pain in the social environment were included. The subsequent stepwise multiple regression analysis demonstrated a significant regression effect of pain in the social environment (*F*(1,25) = 15.743, *p* < 0.001) with an *R*^2^ of 0.362. Including age (*β* = 0.253, *t* = 1.608, *p* = 0.121) or chronic pain status (*β* = 0.227, *t* = 1.352, *p* = 0.189) did not increase *R*^2^.**Biological subscale:** Age (*r* = −0.015, *p* = 0.942), chronic pain status (*r* = 0.127, *p* = 0.527), and pain in the social environment (*r* = 0.314, *p* = 0.111) were not significantly related to the biological subscale.**Psychological subscale:** Pain in the environment accounted for a significant amount of variance in the psychological score (*F*(1,25) = 20.784, *p* < 0.001), with an *R*^2^ of 0.432. In step 2, we found a significant regression effect of age (*F*(2,24) = 14.596, *p* < 0.001), with an *R*^2^ = 0.511 (increase in *R^2^* = 0.095, *p* = 0.034). Including chronic pain status (*β* = 0.116, *t* = 0.672, *p* = 0.508) did not increase *R*^2^.**Social subscale:** Pain in the environment accounted for significant variance in the social score (*F*(1,25) = 6.542, *p* = 0.017), with an *R^2^* of 0.176. Including age (*β* = 0.345, *t* = 1.975, *p* = 0.060) or chronic pain status (*β* = 0.273, *t* = 1.437, *p* = 0.164) did not increase R^2^.Appendix E shows all regression analyses in detail.

## 4. Discussion

The present multimethod study presents a new tool, the biopsychosocial pain concept matrix (BiPS matrix), which assesses children’s pain concepts. This study aimed to pretest this new instrument by (1) assessing children’s reading problems, difficulties in explaining meaning, and unknown words by use of cognitive interviews and (2) testing it within an online sample to gain an initial, insight into modulating factors of a child’s concept of pain such as age, chronic pain status, prior pain knowledge and pain in the social environment.

Findings of the **cognitive interviews** in nine children revealed that reading problems occurred primarily in items with unknown words. Problems explaining the meaning occurred in 65% of items and co-occurred with unknown words. Understandability of items was acceptable to good, but results indicated the need for further adaptation of the wording. This result is in accordance with the literature showing that adaptation of questionnaires for use in children is key [34]. Some children explained items from the perspective of their real-life experiences, leading to unintended interpretations of the item texts. Another interesting finding related to how children understood adult concepts of pain differently (Table 1). Of interest, the construct of guilt was not mentioned in the cognitive interviews even though previous studies revealed that guilt seems to play an important role in a child’s concept of pain [35,36]. One explanation for children in this sample not talking about guilt could be that they were asked about the understandability of items while having to focus on their own experiences with pain.

Children in the **online study** (*N* = 27) reported good comprehension of items of the BiPS matrix. The same version of the questionnaire was used in this study. Looking at the psychometric characteristics, the BiPS total score showed high internal consistency of items and moderate difficulty. Difficulty is acceptable on all subscales. In line with the heterogeneous nature of conceptual knowledge and our expectations, the internal consistencies of some scales were low (for timeline and cause) or questionable (social, consequences, type of pain), even though they were higher for others (biological, psychological, treatment). This points to the importance of future studies investigating how psychoeducation can influence the fragmentation and integration of children’s pain concepts [37].

Correlational analyses preceding further analysis of modulating factors yielded several findings: age, chronic pain status, and pain in the social environment correlated with children’s pain concepts, whereas prior pain knowledge was unrelated to the BiPS total score and all subscales. 

Regression analyses showed that pain in the social environment was positively related to the BiPS total score and the social subscale. In contrast, age and the children’s own chronic pain status did not account for additional variance in both cases: children who knew people suffering from chronic pain in their social environment and engaged in the topic showed *higher* scores in the BiPS matrix and therefore displayed pain concepts that were more in line with contemporary pain science. We found similar effects on the psychological subscale: pain in the social environment was again positively related to the psychological subscale but also with age, explaining the additional variance. The biological subscale was not explained by any of the predictors.

Our findings suggest that pain in the environment explains the most variance in pain concepts of children. This new finding indicates that being in contact with people suffering from chronic pain might lead to a pain concept more in line with contemporary pain science. However, it is important to note that we recruited children from a selective sample where one of their parents had cancer.

Furthermore, children with chronic pain reached higher mean scores in the BiPS matrix than those without pain. However, due to the small sample size, these findings must be interpreted with caution. This finding contrasts with Pate et al. (2020) [28], who found a negative correlation between mean scores in the COPI (pain concepts) and pain intensity in children. This might be justified through the different sample compositions: while Pate et al. [28] assessed a clinical sample, we asked supposedly healthy children of parents with cancer. Moreover, Pate et al. (2020) [28] assessed pain status and pain intensity (using an 11-point numeric rating scale) while we did not assess pain intensity. We solely assessed chronic pain status (rating “*yes*” or “*no*”).

What might be considered an astonishing finding is that children’s prior pain knowledge was unrelated to the BiPS matrix total score or any subscales. One reason could be that children were asked to self-assess prior knowledge and might not have been able to estimate it correctly. Self-assessment should be critically discussed as a suitable approach to assessing prior pain knowledge. Another striking finding was that none of our predictors were significantly associated with the biological subscale. One possible explanation for this might be that our chosen predictors (chronic pain status, prior pain knowledge, pain in the social environment) simply are not suitable to predict variance on the biological subscale. Maybe different factors such as school grades in biology could account for variance on the biological scale. Parental effects should also be taken under consideration: since the children in our sample all had at least one parent with cancer, they might have already gained more knowledge of the psychological and social aspects regarding chronic pain while biological aspects are not as evident in everyday life. Age was also not significantly related with the biological subscale. It seems imaginable that biological items of the BiPS matrix were easy to understand so that older children did not have an advantage over younger children.

### Limitations

While this study aims to test a multidimensional tool for assessing pain concepts in children, our findings must be interpreted in view of some methodological limitations: Results from both cognitive interviews and the online study are based on a small sample size with a wide age range (9 to 19 years). Clearly, future studies are warranted to investigate developmental factors on children’s pain concepts. Furthermore, due to the COVID-19 pandemic, five cognitive interviews had to be performed online via Zoom rather than face-to-face. This might have had an impact on the children’s response behavior. Results concerning the chronic pain status (e.g., comparing children with and without chronic pain) should be interpreted with caution because of a potential bias due to the enrollment of participants at a center for children of parents with cancer. In addition, the sample comprised high-aptitude students, with most children (66.7%) attending the Gymnasium, the highest school track in Germany. Moreover, our participants might have had more prior pain knowledge since children were recruited from a selective sample where one of their parents had cancer. This knowledge could have had an impact on children’s pain concepts and might therefore limit generalization of our results.

We found good comprehension of the BiPS matrix in the online study, which is a contrasting finding to the results of the cognitive interviews where children struggled with explaining the meaning of the items. This result should be interpretated with caution because we assessed understanding of items differently in part 1 (cognitive interviews) and part 2 (online study): while participants had to explain items in part 1 followed by researchers ranking their understanding accordingly, participants in part 2 rated their understanding themselves. Given that both studies are designed to be preliminary, future research into the understanding of a revised version of the BiPS is warranted.

Lastly, the 5-point Likert scale for the BiPS items ranged from “*totally disagree*” to “*totally agree"*, with the middle category being “*I’m not sure*”. We chose agreement levels rather than true and false as answer categories to assess concepts rather than knowledge. However, this might have led children to agree with items (bias).

## 5. Conclusions

The BiPS matrix is a novel tool to assess pain concepts in children by combining the biopsychosocial framework and the domains of the Common-Sense Model of Self-Regulation [14]. The results can inform further improvements of the BiPS matrix, e.g., regarding the wording of items and the shortening of the questionnaire. Findings also deliver the first evidence of the validity of the BiPS matrix for assessing pain concepts as revealed by positive associations with children’s age, chronic pain status, and pain in the social environment. The importance of being in contact with people suffering from chronic pain was demonstrated by the impact that pain in the social environment had on children’s pain concepts. We consider the present project an initial step in the development of a new diagnostic tool. The following research agenda can be proposed based on the present results: testing of the improved and abbreviated BiPS matrix in larger samples of healthy children and children with chronic pain, their parents, and healthcare providers in order to assess their biopsychosocial pain concepts. It is substantially important to disentangle possible knowledge gaps in healthcare providers in order to improve pain therapy.

Prospectively, the BiPS matrix may be used for individualizing pain science education in children with chronic pain based on their pain concept profile. 

## Figures and Tables

**Table 1 children-10-00814-t001:** Examples from cognitive interviews with children.

Item	Original Item	Children’s Understanding
1	“*Chronic pain does not have a warning function.*”	“*Chronic pain is a hereditary disease.*”
4	“*Having suffered from pain for a long time makes the brain more sensitive to warning signals of the body.*”	“*If you break your leg once*, *you shouldn’t do it again.”* *“I was in pain and now I am more careful.*”
7	“*Pain intensity changes depending on what you know about pain.*”	“*If you break your spine and know that you won’t be able to walk any more*, *that can influence your pain.*”
8	“*Pain intensity changes depending upon who you are with.*”	“*It’s always embarrassing to show pain in front of friends*, *no matter how close we are.*”
12	“*You can have an injury and feel no pain.*”	“*For example*, *when feelings are hurt.*”
14	“*Worrying about pain can make you feel more pain.*”	“*When I am in pain*, *I worry.”* *“If my soul is injured and on top of that*, *I worry*, *then I feel even worse.*”
20	“*Too many pain killers can cause persisting pain.*”	“*Medication can have side effects.”* *“Taking medication means that the pain is so bad you can’t take it anymore.*”
23	“*Having chronic pain can change your everyday life.*”	“*Chronic means it is written in your thoughts.*”
25	“*Having chronic pain can make you want to meet up with your friends less often.*”	“*Does chronic mean that it is something in your head?*”
28	“*You can suffer from chronic pain even after an injury is healed.*”	“*Even after an injury is healed*, *the scare can still hurt.”* *“This only happens when an injury heals the wrong way.*”
29	“*Being happy can make you feel less pain.*”	“*If you want to hang out with friends really badly*, *one can suppress the pain.*”
33	“*It is not always necessary to take medication when dealing with chronic pain.*”	“*Medication isn’t effective when pain is associated with the soul.*”
38	“*When doctors*, *therapists*, *and nurses work together*, *they can help reduce pain intensity.*”	“*If doctors surprised me on my birthday or something*, *I would be better.*”

All items originally in German.

**Table 2 children-10-00814-t002:** Psychometric properties of the vertical and horizontal dimensions of the BiPS matrix after elimination of nine items (*N* = 31 items).

Dimension	*k* (Number of Items)	*R_it_* (Item-Total Correlation)	*P* (Difficulty Index)	*ɑ* (Cronbachs Alpha)	*M* (BiPS Matrix Total Score)	*SD* (Standard Deviation)
Biological	9	0.41	69.78	0.73	2.63	0.33
Psychological	12	0.44	71.48	0.78	2.85	0.42
Social	10	0.30	68.18	0.63	2.61	0.32
Cause	7	0.28	62.50	0.49	2.31	0.38
Consequences	7	0.33	72.93	0.61	2.84	0.40
Type of pain	6	0.39	73.38	0.65	2.85	0.45
Timeline	3	0.27	71.17	0.42	2.78	0.33
Treatment	8	0.41	71.38	0.71	2.82	0.45

*Notes*. B = biological; C = assumptions about cause; Con = consequences of pain for sufferers; *M* = mean; *MAX* = maximum; *MIN* = minimum; *P* = percent difficulty index; P = psychological; *r_it_* (*BiPS*) = corrected item scale correlation for the biopsychosocial scales; *r_it_* (*content*) = corrected item scale correlation for the content dimensions; S = social; *SD* = standard deviation; Td = type of disorder; Time = time course of disease; Treat = possibility of control and treatment. All values were rounded to two decimal places.

## Data Availability

The authors are not able to make the data and scripts of the project available, given that consent for data sharing was not obtained at the beginning of the study in 2020.

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
