# Peer review of "Know Pain, No Pain? Preliminary Testing and Application of a New Tool to Assess Biopsychosocial Pain Concepts in Children"

_children, 2023, doi:10.3390/children10050814_

Round 1

Reviewer 1 Report

Important issue, ambitious aim, may be too emphasis on the fact that pain comprehension while relevant is one of the many perpetuating factors of chronic pain.

Methods : it is not clear if the tool tested online was modified according to the cognitive interview results .

Online cognitive interview sample is a possible limit to be discussed , how many interview  online and how many in person ? .

How was the range 9-19 chose : there is a huge difference between 9 and 19 .

The high the level of instruction of the population is properly  reported as a possible limit. Moreover children with learning disabilities may experience more somatic symptom disorders due to a disproportion  between their performance and the environmental requests.  Was there any exclusion criteria  ( eg children with learning difficulties, ADHD .. )?

Please explain why enrollment took place  at a center for children with parents with cancer ? Isn’it a possible bias ? This issue is eventually tackled in the discussion section being acknowledged as a limit for generalization of the results. However the limit section should be strengthened by adding the limited number of children enrolled and the wide age variation 9-19.

Part 1 . Cognitive interview . Convenience sample, of 9 children for cognitive interview . Seems a limited number.

How was the “convenience sample” defined ? Previous adults’ or childrens’ studies ?

The item “ pain in the social environment (Do you  know anyone suffering from chronic pain? on a 5-Point Likert Scale ranging from no to yes,  and I engage in this topic)” may mask a possible family bias , as well as the selected population ( children of parents with cancer) .

52% of reading problem is quite high, 60% of explaining meaning struggle is high as well, The percentage of making sense items ( 42% ) is also very low : at the light of these results how can results of the online study be considered so good ? Please comment on this.

Part 2, online study .

Again : how was the population selected ?

30% of selected patients had chronic pain, some more information about this population in a supplementary material would be helpful.

In any case 30% does not represent the standard population , why was this percentage chosen ?

It is not clear from the text  if the tool tested online was modified according to the cognitive interview results. If this was not the case this should be made clear in the methods and discussion. In this case the perception of the difference of quality of the tool between the two population is quite different according to the presented results. The authors should comment on this issue.

Reviewer 2 Report

This is an exciting study on the multidimensional assessment of pain in Children. It is showing how difficult it is to assess pain when we do not know the child’s concepts of pain. Although this tool is for the self-assessment of pain by children this time it was not used in this sense. The children had limited experience with their own pain, and only few have had experience with the pain of their parents. Although the concept of the soul is mentioned several times the spiritual dimension was not considered here. The children’s common concept of pain and guilt is surprisingly not elaborated here. Most children associate pain with a guilty feeling they did something wrong (and this is why mum has pain. All because of me. All because of is did or did not). I found most discovering in the article how children do understand our adult concepts (table 1.)

In summary I found this paper interesting but not complete. May be further investigations on larger number of patients will bring more. It is worth to try.

Round 2

Reviewer 1 Report

Thanks for the changes, paper is improved.